# Content Determination and Chemical Clustering Analysis of Tanshinone and Salvianolic Acid in *Salvia* spp.

**DOI:** 10.3390/metabo14080441

**Published:** 2024-08-08

**Authors:** Feiyan Wang, Yufeng Bao, Furui Yang, Lu Yuan, Xinchun Han, Yanbo Huang, Yukun Wei, Lei Zhang, Zongqi Yang, Dongfeng Yang

**Affiliations:** 1Key Laboratory of Plant Secondary Metabolism and Regulation in Zhejiang Province, College of Life Sciences and Medicine, Zhejiang Sci-Tech University, Hangzhou 310018, China; 2Eastern China Conservation Centre for Wild Endangered Plant Resources, Shanghai Chenshan Botanical Garden, Shanghai 200120, China; 3Shanghai Botanical Garden/Shanghai Engineering Research Centre of Sustainable Plant Innovation, Shanghai 201600, China; 4Zhejiang Engineering Research Centre for the Development Technology of Medicinal and Edible Homologous Health Food, Shaoxing Biomedical Research Institute of Zhejiang Sci-Tech University Co., Ltd., Shaoxing 312075, China

**Keywords:** *Salvia* spp., tanshinone, phenolic acid, high-performance liquid chromatography

## Abstract

*Salvia miltiorrhiza* is one of the famous traditional Chinese medicines for treating cardiovascular and cerebrovascular diseases. Tanshinone and phenolic acids are the main active compounds of *Salvia miltiorrhiza*, whereas the distribution patterns of the two kinds of components are still unclear among *Salvia* spp. In this work, high-performance liquid chromatography was applied to analyze the distribution patterns of major components in the roots and leaves of 58 *Salvia* spp. The results showed that the distribution patterns of tanshinone and phenolic acids in *Salvia* spp. varied significantly. Phenolic acid components such as rosmarinus acid, caffeic acid, and danshensu are widely distributed in the roots and leaves, and phenolic acids in the leaves of *Salvia* spp. are generally higher than that in roots. Tanshinones are mainly detected in the roots of *Salvia przewalskii*, *Salvia trijuga*, *Salvia castanea*, and *Salvia yunnanensis*. The content of major components of the different species varied significantly. The content of phenolic acids in most *Salvia* spp. generally followed the pattern of salvianolic acid B > rosmarinic acid > danshensu > caffeic acid both in the roots and leaves. Tanshinone IIA and cryptotanshinone were the main lipophilic components of *Salvia* spp. distributed in southwest China. A correlation between the distribution pattern of tanshinone and the genetic relationship of species was indicated in the work. This research systematically reveals the distribution patterns of tanshinone and phenolic acids in *Salvia* spp., providing a theoretical basis for the development and utilization of medicinal resources of *Salvia*.

## 1. Introduction

*Salvia* spp. is an important medicinal plant group, with about 1000 species widely distributed around the world [1,2], containing abundant natural products. Currently, about 300 components have been isolated from *Salvia* spp. and are distributed in China [3]. The components mainly include diterpenoids and phenolic acids [1,4] and contain ingredients with medicinal values such as antithrombotic [5], antioxidant [6,7,8], anti-tumor [1,7,8], and antiviral [1,9]. The famous traditional Chinese medicine *Salvia miltiorrhiza* is a plant of the *Salvia* spp. in the Lamiaceae family, with the effects of promoting blood circulation, removing blood stasis [4], and reducing swelling and pain [7], which is widely applied to treat cardiovascular diseases in clinical practice [1,8,10]. And it is reported that tanshinones and phenolic acids are the main active compounds of *Salvia miltiorrhiza* [7,11]. With the depletion of *S. miltiorrhiza*, cultivated variety quality, and the content of active compounds, the healthy development of the modern traditional Chinese medicine industry using *S. miltiorrhiza* as raw material is severely constrained [12,13,14,15]. Molecular breeding to create new high-yield varieties with active components and reconstructing the synthesis pathway of secondary metabolites in microbial chassis cells to achieve efficient synthesis and stable utilization of active components is the main method to solve the contradiction between the supply and demand contradiction of Chinese herbal medicine [8,16,17,18]. Currently, about 30 species of *Salvia* spp. are reported to be used in folklore as *S. miltiorrhiza* substitutes [4,8,11,19]. Therefore, it is of great significance to systematically study the distribution of tanshinone and phenolic acids for the development and utilization of medicinal resources of *Salvia* spp.

The secondary metabolites derived from plants are the main source of natural products, which are regulated by the environment and genetic background [20]. *Salvia castanea* is a unique species that grows in Xizang (Tibet) and other high-altitude areas [21,22]. The content of tanshinone II A in the roots of *S. castanea* is reported as much as nine times higher than that of *S. miltiorrhiza* [21]. *Salvia bowleyana* is a perennial herbaceous plant, with an efficacy similar to that of *S. miltiorrhiza*. It reported that the content of phenolic acids in *S. bowleyana* was significantly higher than that of *S. miltiorrhiza*, while the content of tanshinone was significantly lower [23]. Ecological factors affect the quality of medicinal herbals including climate, soil, and microorganisms that regulate the expression of key genes and influence the accumulation of secondary metabolites. For example, the expression level of phenylalanine ammonia-lyase, cinnamate 4-hydroxylase, and 4-coumarate genes was significantly up-regulated under stressful conditions, promoting the biosynthesis of lignin, flavonoids, and phenolic acids to enhance the tolerance of metal stress [24,25,26]. *Salvia* spp. has a wide geographical distribution, mainly distributed in tropical and temperate regions [27] in Europe, Africa, East Asia, and North and South America [28,29]. Currently, the study mainly focuses on the morphology, distribution, and chemical composition of East Asia *Salvia* spp. There are no systematic studies available on the chemical composition of plants distributed abroad [4,19,30,31,32,33,34]. Moreover, the distribution pattern of major components within the genus was relatively less [35,36].

In this work, the contents of 14 major components (tanshinone and phenolic acids) in 58 species were determined by (high-performance liquid chromatography) HPLC. The distribution patterns of major components in different species were analyzed. It was of great significance to provide the theoretical basis for the development and utilization of *Salvia* spp.

## 2. Materials and Methods

### 2.1. Experimental Materials

The 58 species of *Salvia* spp. involved in this research were identified by Professor Wei Yukun from Shanghai Botanical Garden as plants of the Lamiaceae family. After they had been planted in Shanghai Botanical Garden for three months, plants were sampled in July 2019. And plant samples were preserved in the Shanghai Chenshan Botanical Garden Museum. Detailed information on samples is listed in Table 1. The tanshinone standards including 1,2-dihydrotanshinquinone, cryptotanshinone, tanshinone I, tanshinone IIA, and miltirone, and phenolic acids standards including danshensu, protocatechuic acid, protocatechualdehyde, caffeic acid, ferulic acid, salvianolic acids A, rosmarinic acid, lithospermic acid, and salvianolic acids B were purchased from the China Academy of Food and Drug Inspection and Research.

### 2.2. Experimental Methods

#### 2.2.1. Preparation of Standard Solutions

The appropriate amount of standard reference (tanshinone) was weighed and placed in a 10 mL centrifuge tube. Equal volumes of 100% chromatographic acetonitrile were added and ultrasound for 45 min. The appropriate amount of standard reference (salvianolic) was weighed and placed acid in a 10 mL centrifuge tube. Added equal volumes of 50% chromatographic methanol and sonicate for 45 min and placed in a 4 °C refrigerator for later use.

#### 2.2.2. Preparation of Sample Solutions

The samples were divided into leaf and root parts and heated in the drying oven to a constant weight (60 °C), and then the samples were crushed into homogeneous powders, respectively. The powder was filtered using a 40-mesh filter screen. The prepared powder (0.02 g) was dissolved with ultrasonic (53 kHz, 350 W) for 60 min at greenhouse temperature in 1.5 mL of 70% methanol (*v*/*v*). The sample was centrifugally separated (15 min, 12,000 rpm) at room temperature, then the supernatants were collected for sample loading.

#### 2.2.3. Standard Injections

The 200 µL of each tanshinone standard reference was taken and mixed. The 20, 15, 10, and 2 µL of mixed solution was accurately injected. The mixed solution was diluted 10 times to be injected for 10, 2, and 1 µL. In view of the similar structures of salvianolic acid B and salvianolic acid A, salvianolic acid B was separately injected for 20, 10, 5, and 2 µL of 1 mg/mL standard solution of salvianolic acid B, respectively. Then, diluted 10 times and injected 5, 2, and 1 µL, respectively. Then, 100 µL of each phenolic acid standard reference was taken, except for salvianolic acid B, and mixed together to be injected as 20, 15, 10, and 2 µL. Then, 100 μL of mixed phenolic acid solution was diluted 10 times and was injected 10 μL and 2 μL, respectively. The peak area of the standard substance was taken as the vertical axis (Y/µv) and the content of the standard substance as the horizontal axis (X/µg), for drawing the standard curve.

#### 2.2.4. Determination of Major Component Content

The Waters e2695 binary high-performance liquid chromatography was equipped with a water 2998 ultraviolet detector and was applied for data acquisition. The Waters SunFire C18 reverse-phase chromatography column was used to separate the components. The chromatographic conditions were as follows: flow rate was 1 mL/min, column temperature was 30 °C, and injection volume was 20 µL. Water containing 0.02% phosphoric acid (A) and acetonitrile (B) performed as the mobile phase. The gradient elution was 5–20% B over 0–10 min, 20–25% B over 10–15 min, 25–30% B over 15–28 min, 30–40% B over 28–40 min, 40–45% B over 40–45 min, 45–58% B over 45–58 min, 58–60% B over 58–70 min, 60–65% B over 70–80 min, 65–90% B over 80–85 min, 90–90% B over 85–90 min, 90–5% B over 90–95 min, 5–5% over B 95–100 min.

#### 2.2.5. Statistical Analysis of Data

Two software named GraphPad Prism 8 (GraphPad Software, San Diego, CA, USA) and JMP Pro 13 (SAS, Cary, NC, USA) were mainly used for statistical analysis in this work. The GraphPad Prism 8 was mainly used for the calculation of the content of main medicinal substances in 58 species among the *Salvia* spp. The chemical clustering based on the distribution pattern of medicinal substances was performed using JMP Pro 13. The Principal Component Analysis (PCA) and Partial least squares discriminant analysis (PLS-DA) models were established using SMICA 14.0.

## 3. Results

### 3.1. Standard Curve Preparation

The peak area normalization method was used to draw the standard linear equation, and the standard equation is shown in Table 2.

### 3.2. Quantitative Analysis of Phenolic Acids in the Roots of Salvia spp.

The results showed significant differences in the content of different phenolic acids in 58 species. Danshensu, caffeic acid, rosmarinic acid, and salvianolic acid B were widely distributed in *Salvia* spp. While species from China had a higher content of phenolic acid components (except protocatechuic aldehyde and protocatechuic acid). In addition, the contents of rosmarinic acid and salvianolic acid B were significantly higher than other phenolic acid, indicating that rosmarinic acid and salvianolic acid B were the main phenolic acids that achieve efficacy in medicinal plants of the *Salvia* spp. Interestingly, as an upstream substrate, the content of caffeic acid was significantly lower than that of rosmarinic acid and salvianolic acid B (Figure 1). The content of rosmarinic acid in *S. daiguii* was as high as 5.334%, and salvianolic acid B in *S. paramiltiorrhiza* was as high as 11.808%. The content of danshensu in *S. sinica* was as high as 0.172% (Figure 1). The content of salvianolic acid A and protocatechuic acid was relatively low in the genus *Salvia* and can only be detected in a few species. It was indicated in the work that *S. nanchuanensis* was found with the highest content of protocatechuic acid, which was 0.061%, indicating that *S. nanchuanensis* can be used as the monomeric source (Appendix A). Caffeic acid is a heteropantameric substrate that is involved in the synthesis of multiple classes of components [37]. Phenolic acids with a furan structure, such as salvianolic acid B and lithospermic acid, were widely detected in species that are distributed in China but can hardly be detected in species distributed in foreign areas. The results indicated that heterocyclization reactions mainly occur in the roots of *Salvia* spp. Phenolic acids and tanshinones with a furan structure may be the key factors for the unique pharmacological effects of *S. miltiorrhiza*. The specific genetic genes contribute to the uniqueness of the chemical component distribution pattern and lead to the species distribution in China forming an independent branch [38]. The specific metabolites with a furan structure can serve as chemical markers, providing a basis for the phylogenetic relationship of *Salvia* spp.

### 3.3. Quantitative Analysis of Tanshinones in the Roots of Salvia spp.

The contents of major tanshinones in different *Salvia* spp. varied significantly (Figure 2). *Salvia* spp. distributed in the southwest were shown to be a high-yield resource for tanshinone. Tanshinone IIA was highest in the roots of *S. trijuga* (0.685%), which was 4.42 times higher than that in *S. miltiorrhiza*. The content of cryptotanshinone in *S. przewalskii* was significantly higher (0.203%), which was 5.485 times higher than that in *S. miltiorrhiza*. The content of dihydrotanshinone in the roots of *S. miltiorrhiza* was significantly higher, and reached 0.065%, compared with species distributed in the southwest of China. *S. prionitis* has the highest content of miltirone, and it can be used as a high-quality raw material in the large-scale production of miltirone. The species distributed in southwest China were found enriched with tanshinones, whereas the content of phenolic acid was generally lower.

The distribution pattern of effective components in *Salvia* spp. was regulated by external environment and genetic factors [20]. Two synthetic pathways were responsible for the biosynthesis of terpenoids, one was mevalonic acid (MVA) and another was mevalonic (MEP) [39]. The expression level of key enzymes directly determines the content of tanshinone components [40]. The external environment influences the accumulation of effective components of medicinal resources in *Salvia* spp. by influencing the expression of key genes. A low-temperature environment can significantly promote the expression of key genes in the biosynthetic pathway and then upgrade the synthesis of tanshinone II A [41]. According to the pharmacopoeia, tanshinone II A and salvianolic acid B are important majors for evaluating the quality of *S. miltiorrhiza* [42]. The content of tanshinone components in *S. honania*, *S. trijuga*, and *S. przewalskii* was close to the standard in the Chinese Pharmacopoeia; however, the content of salvianolic acid B was significantly lower than that in the pharmacopoeia. The tanshinone components of *Salvia* spp. in southwest China were highly like those in *S. miltiorrhiza*, which provided a basis for further exploration and utilization of *Salvia* spp.

### 3.4. Cluster Analysis of Phenolic Acids in Salvia spp. Roots

The species were mainly divided into three branches based on the distribution pattern of major phenolic acids (Figure 3). The content of salvianolic acid B, rosmarinic acid, and lithospermic acid in the species distributed in central China was relatively high, such as in *S. adiantifolia*, *S. grandifolia*, *S. honania*, *S. meiliensis*, and *S. yunnanensis*, and other southwestern-distributed plants of the *Salvia* spp. *S. deserta*, *S. splendens*, and *S. prionitis*, due to their lower content of salvianolic acid B and rosmarinic acid were grouped together. The low content of danshensu contributed to the clustering of *S. trijuga*, *S. substolonifera*, *S. officinalis*, and some species distributed in foreign areas (Figure 3A). To better demonstrate the distribution of the major active compounds in *Salvia* spp., we conducted a multivariate statistical analysis by constructing PCA and PLS-DA, as shown in Figure 3B. *Salvia* spp. exhibited a clear separation trend to a certain extent. Due to the significant content of rosmarinic acid and salvianolic acid B, *S. miltiorrhiza* and *S. miltiorrhiza Bunge f. alba*, along with other 15 species, were grouped into one branch. The results of PCA showed significant consistency with chemical clustering.

### 3.5. Cluster Analysis of Tanshinones in Roots of Salvia spp.

The *Salvia* spp. was divided into two groups according to the distribution pattern of tanshinone, one group can detect tanshinone, while the other group cannot detect tanshinone (Figure 4). *S*. *Przewalskii* and other species distributed in the southwest of China were gathered into a group along with *S. honania*, *S. meiliensis*, *S. miltiorrhiza*, *S. miltiorrhiza*, and *S. grandifolia*. Tanshinone was obviously detected in these species. The contents of tanshinones were significantly high in *S. miltiorrhiza*, *S. honania*, *S. przewalskii*, and *S. trijuga*. The second group mainly consisted of *Salvia* spp. distributed abroad and some *Salvia* spp. distributed in China, due to the low content of tanshinones components. Those group species generally had high ornamental value. The PLS-DA model suggested that there were significant differences in the distribution of five tanshinone components in *Salvia* spp. The content of tanshinone IIA in *S. przewalskii* and *S. trijuga* was found significantly higher than that in *S. miltiorrhiza* (Figure 4B); however, phenolic acid components were far fewer than those in *S. miltiorrhiza*, especially rosmarinic acid and salvianolic acid B, as shown in Figure 1, indicating that salvianolic acid B and rosmarinic acid are important active natural products that play an important role in the medicinal value of the bulk traditional Chinese medicine *S. miltiorrhiza*.

### 3.6. Cluster Analysis of Major Components in the Roots of Salvia spp.

The chemical cluster analysis was carried out to reveal the distribution of 13 components in the tested *Salvia* spp. The results indicated that the samples can be classified based on geographical distribution (Figure 5A,B). Specifically, the 58 species could be divided into 3 branches. The first branch was about 11 species distributed abroad, with some species distributed in China, such as *S. deserta*, *S. bowleyana*, and *S. coccinea*. The commonality of *Salvia* species in this branch was the low content of salvianolic acid, and the content of some tanshinones was significantly higher. *S. honania*, *S. miltiorrhiza*, *S. trijuga*, and *S. przewalskii* had a significant accumulation of tanshinones and were classified into the second branch. *S. adiantifolia*, *S. grandifolia*, *S. meiliensis*, and *S. cinica* were clustered into the third branch. The species of salvianolic acid in these species were more diverse, and the content was significant. In addition, we found that the distribution of 13 major components in *S. honania* and *S. miltiorrhiza* were obviously consistent, indicating that *S. honania* was a medicinal plant with high application value. Compared to the components of tanshinone, phenolic acid contributed more to the classification of *Salvia* spp., mainly because the content of phenolic acid was relatively high, especially rosmarinic acid and salvianolic acid B (Figure 5B).

The results of chemical clustering showed the classification of *Salvia* spp. based on different major components, which were significantly different. The plants were mainly divided into three branches based on the distribution pattern of major phenolic acids. Rosmarinic acid was higher and widely distributed in *Salvia* spp., the content of trace phenolic acids played an important role in the chemical clustering of the species. Danshensu was widely distributed in the species distributed in central China. Therefore, it was distinguished from other species. Because of the low content of protocatechualdehyde, the plants in the second branch were distinguished from the southwest-distributed *Salvia*. The 58 species can be divided into two main groups according to the distribution of tanshinones, namely, the species that accumulate major tanshinone components, and the species that do not. The distribution pattern of tanshinone components divided the tested species into a group of species distributed in southwest China, and the other group that was distributed abroad and in central China. According to the distribution pattern of 14 major active components, the tested species can basically be classified according to geographical distribution, which was like the clustering results of nine major phenolic acids. The accumulation pattern of the active components of tanshinone in *S. honania* was like that of *S. miltiorrhiza*, with both having a high content of five major components of tanshinone and salvianolic acid B. Plants such as *S. honania* and *S. cinica* were close to *S. miltiorrhiza* in classical taxonomy that belongs to Subg. Sclarea (Moench) Benth, which was consistent with our results [43]. *S. adiantifolia*, *S. filicifolia*, *S. baimaensis*, *S. cavaleriei*, and other *Salvia* were classified into one group, which was consistent with the research results [44].

### 3.7. Quantitative Analysis of Phenolic Acids in Salvia spp. Leaves

The 13 major components in leaf tissues were also analyzed. The results showed that the eight phenolic acids, including tanshensu, protocatechualdehyde, protocatechuic acid, caffeic acid, ferulic acid, isoferulic acid, rosmarinic acid, and salvianolic acid B were detected in the leaves, whereas tanshinones were not detected in leaves of *Salvia* spp. (Figure 6 and Appendix A).

The contents of phenolic acids in the leaves of different species were significantly different. The content of rosmarinic acid in *S. scapiformis* was the highest with up to 13.451%. The content of caffeic acid in *S. substolonifera* was about 0.351%. However, the contents of caffeic acid were relatively lower, such as in *S. splendens* (0.009%), *S. aerea* (0.014%), *S. mekongensis* (0.0195%), and *S. castanea* (0.008%). No caffeic acid was detected in the leaves of *S. plectranthoides*, *S. cavaleriei* (S0994), and *Salvius cyclostegia*. Caffeic acid was mainly used in the synthesis of downstream phenolic acids with complex structures as a precursor compound. Ferulic acid was only detected in the leaves of some *Salvia* spp., among *S. hylocharis*, which had a significant accumulation of ferulic acid, with the content as high as 0.89%, followed by *S. digitaloides*, with the content as high as 0.74%. Salvianolic acid B had a higher content in the leaves of *S. cavaleriei* (S0994) and *S. nanchuanensis*, which are native to China, but it was rarely detected in the leaves of *Salvia* plants abroad. The content of rosmarinic acid and salvianolic acid B in the leaves of the same plant was generally higher than that of the other six major phenolic acid compounds, maybe because salvianolic acid B and rosmarinic acid are the main phenolic acid efficacy substances. The content of phenolic acid in the leaves was generally higher than that in the roots, which indicated that the leaves of *Salvia* spp. may be an ideal material for extracting phenolic acids.

### 3.8. Cluster Analysis of Phenolic Acids in Leaves of Salvia spp.

The distribution pattern of phenolic acids in *Salvia* leaves showed significant differences and divided the species into four branches (Figure 7A,B). *S. scapiformis*, *S. nanchuanensis*, *S. liguliloba*, and *S. bowleyana* were gathered into one group. The leaves are related to the high content of rosmarinic acid, salvianolic acid B, and tanshensu. *S. hylocharis*, *S. digitaloides*, and *S. flava* were close relatives. *S. trijuga*, *S. substolonifera*, *S. daiguii*, and *S. nanchuanensis* were grouped into a new group. Rosmarinic acid and ferulic acid significantly accumulated in the leaves, but salvianolic acid B could not be detected. *S. nanchuanensis*, *S. miltiorrhiza*, *S. miltiorrhiza*, and *S. nipponica* were divided into one group. In line with previous studies, we found that *S. trijuga* was closely related to *S. substolonifera*, while other species that were distributed in China and foreign areas were gathered into one group. This was quite different from the results of previous studies on the evolution of the species, indicating that in the long-term evolution process, the leaves of *Salvia* spp. were greatly affected by the environment. The classification of *Salvia* in the southwest distribution indicated that the genetic factors of *Salvia* in the southwest distribution have differentiated to adapt to the special climatic conditions at high altitude.

## 4. Conclusions

*Salvia* spp. is rich in medicinal resources and has great medicinal value. Due to the wide geographical distribution, the valuable plants in the genera have not yet been fully excavated. In this study, the distribution of salvianolic acids and tanshinones in the 58 species of *Salvia* spp. was systematically analyzed. The significant differences in the contents of major components were found in different species. Phenolic acids were widely distributed in the roots and leaves of *Salvia* spp. Tanshinones were mainly distributed in roots of the southwestern *Salvia* spp., but not detected in leaves. In addition, tanshinones were not detected in the roots and leaves of *Salvia* spp. distributed abroad. The contents of the active compounds were of great significance for the resource development and clinical application of pharmacodynamic substances to *Salvia* spp. Follow-up research can be focused on improving the accumulation of pharmacodynamic components, increasing the quality of *S. miltiorrhiza* and its related species, and solving the demand for *S. miltiorrhiza* by means of molecular design breeding and synthetic biology.

## Figures and Tables

**Figure 1 metabolites-14-00441-f001:**
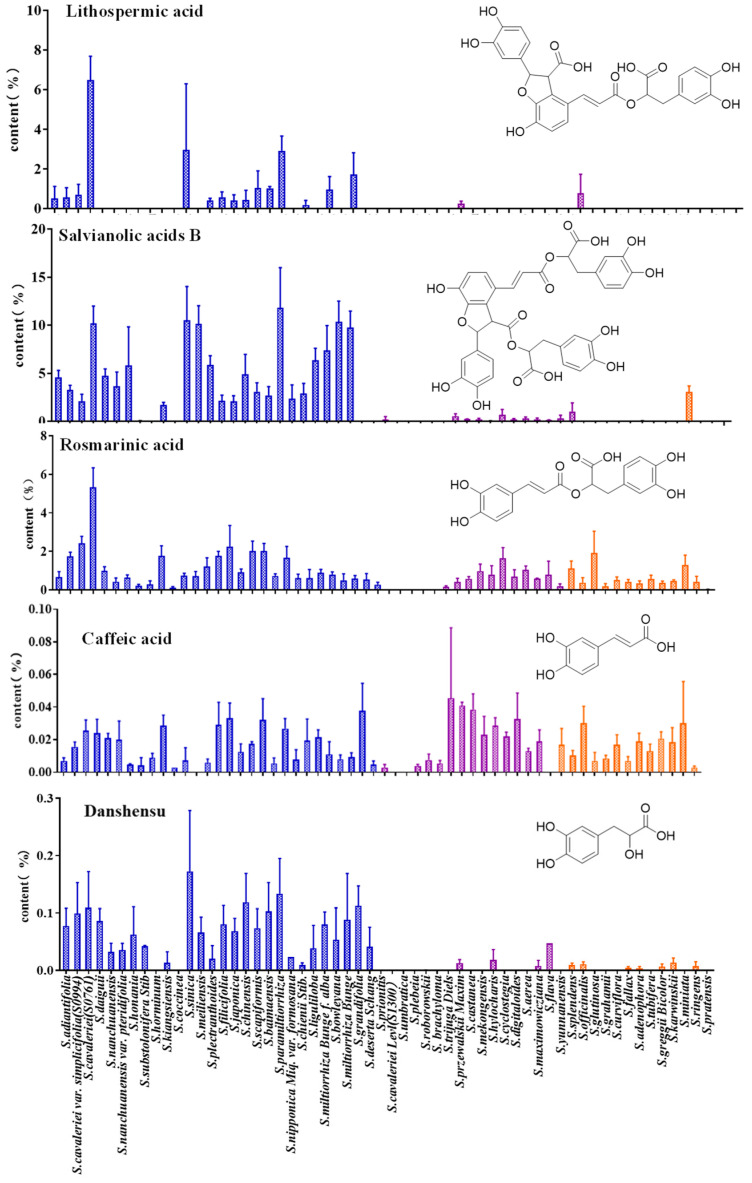
The content of phenolic acids in the roots of *Salvia* spp. Note: The different colors represent species from different sources, Blue represents the regions of China except for Southwest China, Purple represents the southwestern region of China, Orange represents species collected from abroad.

**Figure 2 metabolites-14-00441-f002:**
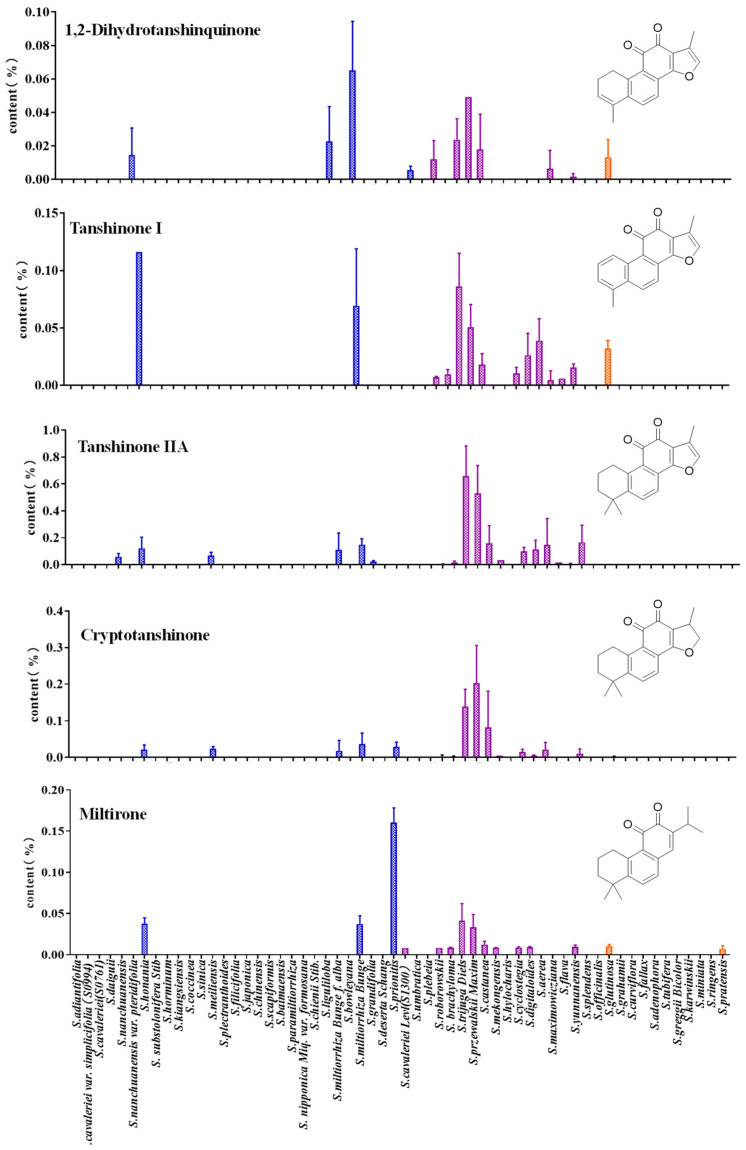
The content of tanshinones in the leaves of *Salvia* spp. Note: The different colors represent species from different sources, Blue represents the regions of China except for Southwest China, Purple represents the southwestern region of China, Orange represents species collected from abroad.

**Figure 3 metabolites-14-00441-f003:**
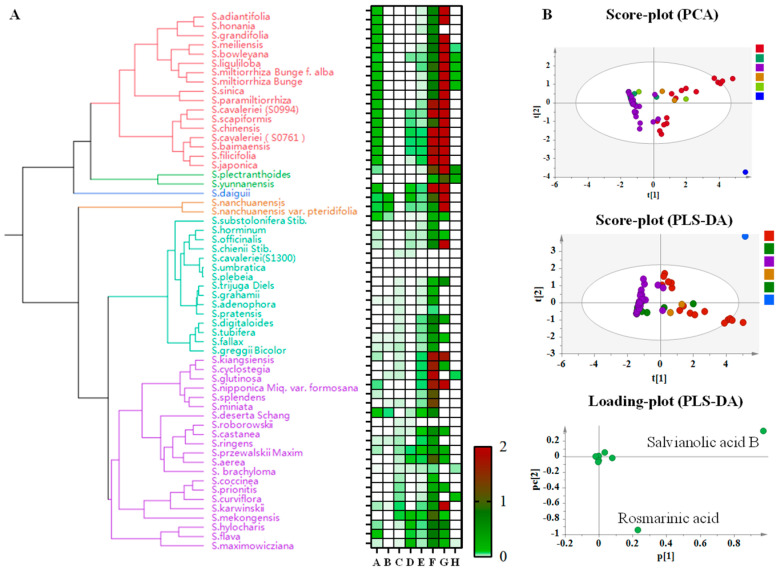
Cluster analysis of phenolic acids in the roots of *Salvia* spp. Note: A: Danshensu, B: Protocatechuic acid, C: Protocatechualdehyde, D: Ferulic acid, E: Caffeic acid, F: Rosmarinic acid, G: Salvianolic acid B, H: Salvianolic acid A. (**A**): Species Evolutionary Tree Analysis Based on Target Component Content. (**B**): Represent multivariate statistical analysis based on target component content. Note: The different colors mean the different species, the red one mean the species same as the species in the evolutionary tree.

**Figure 4 metabolites-14-00441-f004:**
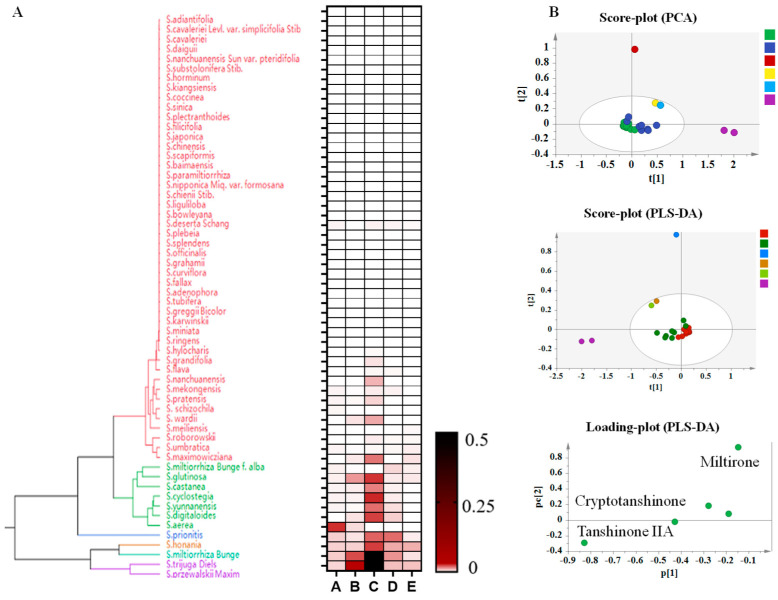
Cluster analysis of tanshinones in roots of *Salvia* spp.: A: Miltirone, B: Cryptotanshinone, C: Tanshinone IIA, D: Tanshinone I, E: Dihydrotanshinone. (**A**): Species Evolutionary Tree Analysis Based on Target Component Content. (**B**): Represent multivariate statistical analysis based on target component content. Note: The different colors mean the different species, the red one mean the species same as the species in the evolutionary tree.

**Figure 5 metabolites-14-00441-f005:**
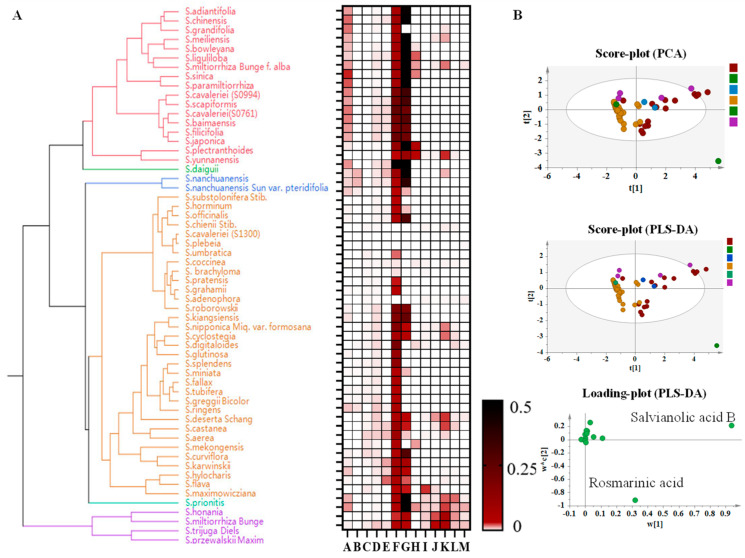
Clustering analysis of major components in the roots of *Salvia* spp.: A: Danshensu, B: Protocatechuic acid, C: Protocatechualdehyde, D: Ferulic acid, E: Caffeic acid, F: Rosmarinic acid, G: Salvianolic acid B, H: Salvianolic acid A, I: Miltirone, J: Cryptotanshinone, K: Tanshinone IIA, L: Tanshinone I, M: Dihydrotanshinone. (**A**): Species Evolutionary Tree Analysis Based on Target Component Content. (**B**): Represent multivariate statistical analysis based on target component content. Note: The different colors mean the different species, the red one mean the species same as the species in the evolutionary tree.

**Figure 6 metabolites-14-00441-f006:**
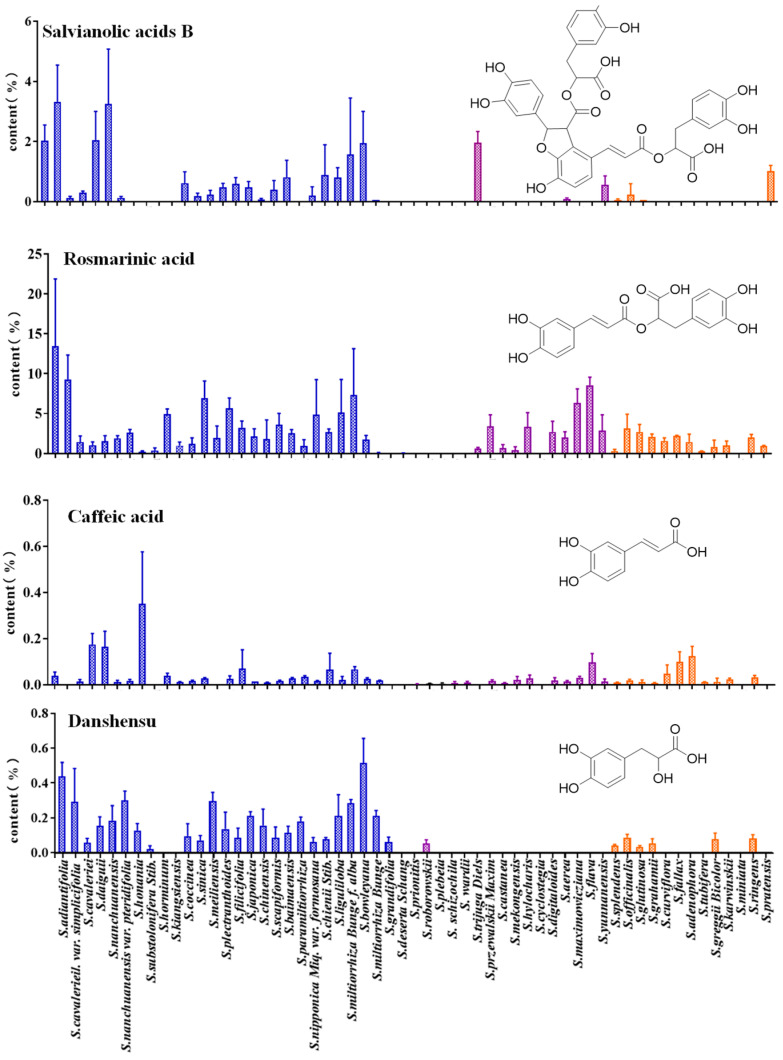
The content of phenolic acids of leaves in *Salvia* spp. Note: The different colors represent species from different sources, Blue represents the regions of China except for Southwest China, Purple represents the southwestern region of China, Orange represents species collected from abroad.

**Figure 7 metabolites-14-00441-f007:**
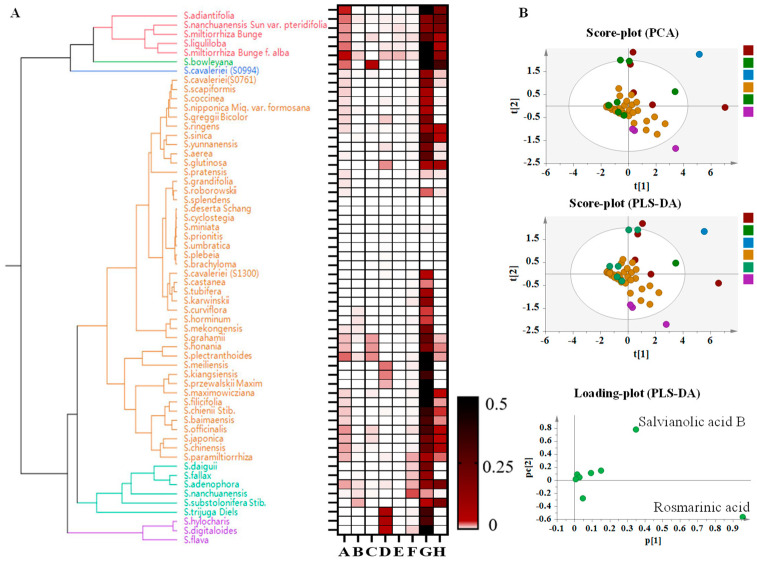
Cluster analysis of phenolic acids in leaves of *Salvia* spp.: A: Danshensu, B: Protocatechuic acid, C: Protocatechualdehyde, D: Ferulic acid, E: Isoferulic acid, F: Caffeic acid, G: Rosmarinic acid, H: Salvianolic acid B. (**A**): Species Evolutionary Tree Analysis Based on Target Component Content. (**B**): Represent multivariate statistical analysis based on target component content. Note: The different colors mean the different species, the red one mean the species same as the species in the evolutionary tree.

**Table 1 metabolites-14-00441-t001:** The samples used in this experiment.

Serial Number	Number	Species (Scientific Name)
1	S0123	*Salvia scapiformis*
2	S0146	*Salvia miltiorrhiza*
3	S0171	*Salvia chinensis*
4	S0175	*Salvia sinica*
5	S0271	*Salvia meiliensis*
6	S0297	*Salvia daiguii.*
7	S0304	*Salvia japonica*
8	S0348	*Salvia baimaensis*
9	S0356	*Salvia honania*
10	S0362	*Salvia miltiorrhiza*
11	S0460	*Salvia plectranthoides*
12	S0484	*Salvia nanchuanensis*
13	S0491	*Salvia paramiltiorrhiza*
14	S0517	*Salvia nanchuanensis*
15	S0603	*Salvia bowleyana*
16	S0671	*Salvia adiantifolia*
17	S0761	*Salvia cavaleriei*
18	S0804	*Salvia grandifolia*
19	S0940	*Salvia filicifolia*
20	S0994	*Salvia cavaleriei*
21	S0477	*Salvia substolonifera*
22	S0742	*Salvia nipponica*
23	S0747	*Salvia liguliloba*
24	S0837	*Salvia deserta*
25	S1094	*Salvia chienii*
26	S1127	*Salvia kiangsiensis*
27	S0318	*Salvia coccinea*
28	S1097	*Salvia prionitis*
29	S1044	*Salvia horminum*
30	S1002	*Salvia officinalis*
31	S1062	*Salvia splendens*
32	S1031	*Salvia glutinosa*
33	S1073	*Salvia grahamii*
34	S1071	*Salvia curviflora*
35	S1072	*Salvia fallax (Salvia roscida)*
36	S1068	*Salvia adenophora*
37	S1082	*Salvia tubifera*
38	S1074	*Salvia greggii*
39	S1089	*Salvia karwinskii*
40	S1078	*Salvia miniata*
41	S1030	*Salvia ringens*
42	S1032	*Salvia pratensis (meadow clary)*
43	S1174	*Salvia maximowicziana*
44	S1153	*Salvia flava*
45	S1151	*Salvia yunnanensis*
46	S1164	*Salvia trijuga*
47	S1170	*Salvia aerea*
48	S1161	*Salvia przewalskii*
49	S1169	*Salvia castanea*
50	S1167	*Salvia mekongensis*
51	S1155	*Salvia hylocharis*
52	S1171	*Salvia cyclostegia*
53	S1173	*Salvia digitaloides*
54	S0421	*Salvia umbratica*
55	S1329	*Salvia plebeia*
56	S1233	*Salvia roborowskii*
57	S1162	*Salvia brachyloma*
58	S1300	*Salvia cavaleriei*

**Table 2 metabolites-14-00441-t002:** Standard curve of Standard references.

Standards	Standard Equations	R2	Linear Range (µg)
Danshensu	Y = 456,192.9651X + 8898.3984	0.999	0.02~2
Protocatechualdehyde	Y = 3,417,750.4961X − 51,858.7541	0.998	0.04~4
Protocatechuic Acid	Y = 2,124,173.3198X + 14,233.8764	1	0.02~2
caffeic acid	Y = 5,472,992.9477X − 3794.4223	1	0.02~2
Ferulic Acid	Y = 3,813,787.9790X − 4677.0069	0.999	0.02~2
Isoferulic Acid	Y = 3,145,231.4942X + 9488.3242	1	0.02~2
Rosmarinic acid	Y = 1,939,973.1010X + 239.4423	0.999	0.02~2
Salvianolic Acids A	Y = 2,115,905.1152X + 7393.8267	0.999	0.02~2
Salvianolic Acids B	Y = 960,469X − 63,957	0.999	0.10~20
lithospermic acid	Y = 111,687X − 8406.9	0.998	0.04~4
Miltirone	Y = 6,351,910.6757X − 181,582.1524	0.994	0.03~3
Cryptotanshinone	Y = 55,262,695.8020X + 85,722.7420	0.999	0.04~4
Tanshinone II A	Y = 5,128,785.8760X − 58,317.4739	0.999	0.04~4
Tanshinone I	Y = 3,724,065.1501X − 86,580.8597	0.999	0.04~4
Dihydrotanshinone I	Y = 2,660,639.5600X + 23,027.7527	0.999	0.04~4

## Data Availability

The original contributions presented in the study are included in the article/Appendix A, further inquiries can be directed to the corresponding author.

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
