# Peer review of "Content Determination and Chemical Clustering Analysis of Tanshinone and Salvianolic Acid in Salvia spp."

_metabolites, 2024, doi:10.3390/metabo14080441_

Round 1

Reviewer 1 Report

Comments and Suggestions for Authors

he authors aimed to systematically evaluate the phytochemical fingerprint (phenolic acids, benzofurans and terpenoid derivates; n=14) of 58 Salvia spp. accessions (roots & leaves) by RF-HPLC and analyzed by cluster/heatmap analysis. All root samples were particularly rich in Danshensu (Salvianic acid A) and coffeic acid< rosmarinic acid and Salvianolic acid B. Although the experimental design seems to be well executed, and both discussion and conclusions unbiased (evidence-based), methods and discussion sections should be modified to improve the manuscript´s uniqueness and scientific soundness. Please consider the following:

General.  A) The manuscript´s reading and comprehension could be improved if it is reviewed by a native English speaker or by a formal translation agency. B) The meaning of each abbreviation must be established when used for the first time (in both main text and table/figures). C) Formatting: Effective paragraphs (https://purdueglobalwriting.center/how-to-write-an-effective-paragraph/) of 10-15 lines are strongly suggested. D) This reviewer is unsure of your use of the phrase indicator, please change or delete.

Abstract. It is somewhat confusing partly because of the incorrect use of language. It is suggested to the authors: A) Change the anecdotal language for specific based assertions (e.g. roots/leaves were rich in Danshensu (Salvianic acid A) and coffee acid < rosmarinic acid and Salvianolic acid ). B) It is advisable to sort by family of chemical structure (phenolic compounds vs. others) denoting the differences between roots and leaves. C) Eliminate HPLC

Introduction. The uniqueness and scientific contribution of the manuscript must be reflected in the last paragraph.

Methods. A) Include and describe in detail the statistical analysis section. B) Anticipating that the authors have many indicators (response variables) for the same experimental treatments, it is advisable to also use a principal components analysis where diversity or communality can be evidenced.

Results & discussion. A) It is requested that the authors make a more inductive than descriptive discussion, using studies in this matter as a basis and expressing hypotheses of use (pharmacognosy) of each variety/group of plants.

Figures/Tables. A) The resolution and sharpness of all figures should be improved (> 300 dpi) and provided with a shorter size when needed. B) It is recommended to leave the strictly necessary figures and transfer the rest (that is, those barely commented on) to supplementary material.

References. A) Modify the format of certain references according to authors ‘guidelines.

Comments on the Quality of English Language

Extensive editing of English language required

Author Response

1、General.  A) The manuscript´s reading and comprehension could be improved if it is reviewed by a native English speaker or by a formal translation agency. B) The meaning of each abbreviation must be established when used for the first time (in both main text and table/figures). C) Formatting: Effective paragraphs (https://purdueglobalwriting.center/how-to-write-an-effective-paragraph/) of 10-15 lines are strongly suggested. D) This reviewer is unsure of your use of the phrase indicator, please change or delete.

Reply: Thank you very much for your revision suggestions. The phrase “indicator” has been changed to “major” and would like to express our sincere gratitude once again.

2、It is somewhat confusing partly because of the incorrect use of language. It is suggested to the authors: A) Change the anecdotal language for specific based assertions (e.g. roots/leaves were rich in Danshensu (Salvianic acid A) and coffee acid < rosmarinic acid and Salvianolic acid). B) It is advisable to sort by family of chemical structure (phenolic compounds vs. others) denoting the differences between roots and leaves. C) Eliminate HPLC

Reply: Thank you very much for your revision suggestions, we have revised and improved the abstract, and HPLC has been eliminated

3、Introduction. The uniqueness and scientific contribution of the manuscript must be reflected in the last paragraph.

Methods. A) Include and describe in detail the statistical analysis section. B) Anticipating that the authors have many indicators (response variables) for the same experimental treatments, it is advisable to also use a principal components analysis where diversity or communality can be evidenced.

Reply: Thank you very much for your suggestion. The methods of statistical analysis were added in the manuscript. And we highly appreciate your second suggestion, unfortunately, due to the fact that the manuscript mainly focuses on of the distribution patterns of specific components in more than 50 species of Salvia species, the peak extraction was not performed, so it is difficult for us to perform the principal component analysis.

4、Results & discussion. A) It is requested that the authors make a more inductive than descriptive discussion, using studies in this matter as a basis and expressing hypotheses of use (pharmacognosy) of each variety/group of plants.

Reply: Thank you for your suggestion, it is very helpful for improving the quality of the manuscript. A more inductive than descriptive discussion was revised in the manuscript.

5、Figures/Tables. A) The resolution and sharpness of all figures should be improved (> 300 dpi) and provided with a shorter size when needed. B) It is recommended to leave the strictly necessary figures and transfer the rest (that is, those barely commented on) to supplementary material.

Reply: Thank you very much. We have made corresponding modifications based on your suggestions.

6、A) Modify the format of certain references according to authors ‘guidelines.

Reply: Thank you very much for giving the opportunity to revise the manuscript. The reference format has been revised according to authors ‘guidelines.

Reviewer 2 Report

Comments and Suggestions for Authors

The manuscript «Content determination and chemical clustering analysis of tanshinone and salvianolic acid in genus Salvia» is devoted to the investigation of the distribution patterns of tanshinone and phenolic acids in Salvia species, providing a theoretical basis for the development and utilization of medicinal resources of Salvia.

The paper needs to be revised before publishing:

  1.  Detection limits and quantification limits for the analyzed compound should be determined.
  2. The authors should provide the temperature of the drying oven (line 105).
  3.  The overall quality of figures should be improved.
  4. Some grammer mistakes occur in the text. Not being a native English speaker, I do not make any remarks on the language. Still, I would recommend to have the manuscript proofread.

Author Response

1、Detection limits and quantification limits for the analyzed compound should be determined.

Reply: I am very sorry, the detection limits and quantification limits for the analyzed compound was added in the table 2 in the manuscript.

2、The authors should provide the temperature of the drying oven (line 105).

Reply: Thank you very much for your revision suggestions. We have made corresponding revisions in the text and highlighted them in red.

3、 The overall quality of figures should be improved.

Reply: Thank you very much for your revision suggestions. The overall quality of figures has been improved.

4、Some grammar mistakes occur in the text. Not being a native English speaker, I do not make any remarks on the language. Still, I would recommend having the manuscript proofread.

Reply: We have made modifications to the language of the manuscript and highlighted it in red.

Round 2

Reviewer 1 Report

Comments and Suggestions for Authors

I appreciate your efforts to partially address my suggestions to improve your manuscript. However, this reviewer considers that there is still room to improve the v2-manuscript:

1) You should use "Salvia spp." instead of "genus Salvia" from title onwards.

2) If you were able to make heatmaps with individual phytochemical composition (Fig 3-5, 7), then a principal component analysis can be performed with this information as well. It is advisable to include PCA to easily identify Salvia species richest in bioactives.

3) Common English ("...the standard equation(s) was (are) shown in Table 2"") and technical English (e.g. "coffeic acid", Caffeic acid) did not improve. It is requested that the manuscript be sent to an agency formal translation and send the certificate in the next delivery.

Comments on the Quality of English Language

Extensive editing of English language required

Author Response

1、You should use "Salvia spp." instead of "genus Salvia" from title onwards.

Reply: Thank you very much for your revision suggestions. We have made corresponding revisions in the manuscript and highlighted them in red.

2、If you were able to make heatmaps with individual phytochemical composition (Fig 3-5, 7), then a principal component analysis can be performed with this information as well. It is advisable to include PCA to easily identify Salvia species richest in bioactives.

Reply: Thank you very much for your revision suggestions. The modifications have been made based on your suggestions. Please refer to the manuscript for details.

3、 Common English ("...the standard equation(s) was (are) shown in Table 2"") and technical English (e.g. "coffeic acid", Caffeic acid) did not improve. It is requested that the manuscript be sent to an agency formal translation and send the certificate in the next delivery

We are very sorry for this small mistake and have checked the manuscript and made revisions for the same error.

Reviewer 2 Report

Comments and Suggestions for Authors

The manuscript has been sufficiently improved. However, the authors still need to correct some grammar mistakes and improve style. For example, the phrase highlighted in red in lines 44-45 should be modified, because it is not quite clear what the authors lmeant here.

Author Response

1、The authors still need to correct some grammar mistakes and improve style. For example, the phrase highlighted in red in lines 44-45 should be modified, because it is not quite clear what the authors meant here.

Reply: Please allow me to apologize for the simple mistakes. We have made corresponding revisions in the manuscript and highlighted them in red.